# An Empirical Study of Multitask Learning to Improve Open Domain Dialogue Systems

**Mehrdad Farahani**[1]    **Richard Johansson**[1,2]

[1]Chalmers University of Technology, [2]University of Gothenburg

{mehrdad.farahani, richajo}@chalmers.se

## Abstract

Autoregressive models used to generate responses in open-domain dialogue systems often struggle to take long-term context into account and to maintain consistency over a dialogue. Previous research in open-domain dialogue generation has shown that the use of *auxiliary tasks* can introduce inductive biases that encourage the model to improve these qualities. However, most previous research has focused on encoder-only or encoder/decoder models, while the use of auxiliary tasks in *decoder-only* autoregressive models is under-explored. This paper describes an investigation where four different auxiliary tasks are added to small and medium-sized GPT-2 models fine-tuned on the PersonaChat and DailyDialog datasets. The results show that the introduction of the new auxiliary tasks leads to small but consistent improvement in evaluations of the investigated models.

## 1 Introduction

In recent years, open-domain dialogue systems have experienced increased research due to the availability of large corpora of dialogue and advances in deep learning techniques (Gao et al., 2018). Unlike task-oriented dialogue systems designed for specific domains or tasks, such as flight booking, hotel reservation, customer service, and technical support (Budzianowski and Vulić, 2019; Budzianowski et al., 2018; Chao and Lane, 2019), open-domain dialogue systems aim to have long-term connections with users by satisfying their emotional, social, and communication needs. Therefore, such a system must comprehend the dialogue context and user demands in order to select the appropriate skill at the appropriate time and generate consistent (Li et al., 2016b; Luan et al., 2017) and grounded (Ghazvininejad et al., 2018; Moon et al., 2019) interpersonal responses. Open-domain dialogue systems can include single-turn or multi-turn

dialogues, where the context and topic of the conversation may change throughout the interaction (Dinan et al., 2020).

Multi-turn open-domain dialogue systems need to maintain the context of the conversation, generate appropriate responses concerning the context and predefined characteristics (persona), and handle various forms of input. A persona is a set of characteristics or attributes that describe a virtual agent's background, personality, and behavior. These attributes include the agent's name, age, gender, profession, interests, and other aspects (Zhang et al., 2018), while a conversation context refers to the background information or previous interactions relevant to the current conversation, with word-level and utterance-level dependencies (Zhao et al., 2020).

Recent developments in transformer-based architectures (Vaswani et al., 2017; Raffel et al., 2020; Lewis et al., 2020) for large-scale pre-training, such as OpenAI's GPT-2 (Radford et al., 2019), have shown exceptional results. Later, the models are fine-tuned via more technical steps and at different scales on a large-scale dialogue corpus (Zhang et al., 2020; Adiwardana et al., 2020; Thoppilan et al., 2022; Shuster et al., 2022).

Pre-trained models on conversational datasets typically process dialogue context (a list of utterances) as a sequence of tokens per utterance to generate responses. Although these approaches show effective results compared to previous approaches, they still need to catch the latent information in more complex structures rather than just tokens (Gu et al., 2021; Zhao et al., 2020). A conversational domain is distinguished by the presence of another component called utterances[1] that plays an imperative role in conveying higher-level information in addition to tokens and their local relationships. Recent research has put forth the use of *auxiliary tasks*

---

[1]An utterance is a spoken or written sentence or phrase that is used to convey meaning or participate in a dialogue.

as a form of regularization during the fine-tuning of models as a means to address the aforementioned issue. A majority of these additional training objectives are implemented solely on the encoder-only and encoder-decoder architectures.

However, while the use of auxiliary tasks has led to improvement in encoder-only and encoder/decoder models, recent work has not explored the application of auxiliary tasks in *decoder-only* models. In this research, we propose incorporating auxiliary tasks on top of an autoregressive decoder-only model to examine and enhance the quality of generated responses concerning the latent information present within utterances. Additionally, we demonstrated the impact of various auxiliary tasks on distinct elements of dialogue across two benchmark datasets. By examining the effect of different auxiliary tasks on various components of dialogue, we aimed to provide a deeper understanding of how these tasks can influence the performance and outcomes of conversational systems. Additionally associated code to this research can be found in our GitHub repository.[2]

## 2 Related Works

The motivation for this research is drawn from recent investigations into the utilization of auxiliary tasks to enhance the generated responses in open-domain dialogue systems by considering the context. To this end, we present and analyze these recent studies in this section. Previous studies in this field can be broadly classified into three general categories. The first category pertains to the widespread use of encoder-decoder models in dialogue response generation, which have been observed to produce generic and uninteresting responses (e.g., "I'm good", "I don't know"). Zhao et al. (2020) proposed an encoder-decoder architecture with two auxiliary tasks at token and utterance levels that can effectively exploit conversation context to generate responses, including order recovery and masked context recovery. Analogously, Mehri et al. (2019) examined a range of unsupervised pre-training objectives for acquiring dialogue context representations via encoder-decoder models by incorporating four auxiliary tasks, including next-utterance retrieval, next-utterance generation, masked-utterance retrieval, and inconsistency identification.

DialogBERT is a unique design that employs a

hierarchical Transformer architecture to comprehensively capture the context of dialogue (Gu et al., 2021). Using two training objectives, similar to BERT (Devlin et al., 2019), allows the model to understand a conversation's nuances effectively. In the first objective, masked context regression, the model is trained to predict the missing context from a dialogue, and in the second objective, distributed utterance order prediction, the model is trained to predict the order of spoken utterances in a conversation so that it understands the flow and logic.

Lastly, decoder-only models, like DialoGPT (Zhang et al., 2020), make use of only the final component of the encoder-decoder structure. DialoGPT in particular, extends the GPT-2 (Radford et al., 2019) architecture by being specifically developed and trained on a large corpus of dialogue data to generate responses in a conversational context. However, despite its ability to perform well in single-turn conversation, its lack of capability to capture latent information behind utterances in a multi-turn conversation, results in an inadequate understanding of the context. The utilization of auxiliary tasks in decoder-only models is a well-established practice. For instance, the GPT-2 based model TransferTransfo (Wolf et al., 2019), which adopts a multi-task objective, showed improvement over the basic GPT-2. These auxiliary tasks primarily take the form of sequence classification tasks.

## 3 Method

### 3.1 A Problem Definition

In this section, the necessary notations utilized are presented, and the learned tasks are briefly outlined. Let $d^{(i)} = (p_1, p_2, \ldots, p_N, u_1, u_2, \ldots, u_T)$ denote the $i$-th dialogue session in the dataset $\mathcal{D}$, where $\mathcal{C} = (u_1, u_2, \ldots, u_{T-1})$ is the dialogue context (history), $\mathcal{P} = (p_1, p_2, \ldots, p_N)$ is the dialogue persona (personality of the system) and $u_T$ is the response regarding to the persona and the context. Each $u_i = \left( w_1^i, w_2^i, \ldots, w_{|u_i|}^i \right)$ in $\mathcal{C}$ is an utterance and $w_j^i$ is the $j$-th word in $u_i$. Then, we aim to generate contextually relevant responses for multi-turn conversations using self-supervised auxiliary tasks. Our approach involves two major components, a language model trained based on the GPT-2 and a classification model on top of the GPT-2 used for auxiliary parts. This simple structure has been found to be effective in producing consistent responses. As such, two auxiliary

[2]https://github.com/m3hrdadfi/MoGPT

tasks have been designed over language modeling (LM) to improve the system's performance further. Order and masked recovery tasks are designed to enhance the self-attention module's capacity to capture linguistic affinities. The utterance permutation task enhances the self-attention module's ability to grasp word and utterance sequences, while the masking task seeks to reinforce semantic connections between words and utterances by optimizing the self-attention mechanism. These auxiliary tasks are critical in providing additional supervision signals to the model, leading to improved language modeling performance. Figure 1 illustrates the model. Lastly, a total loss function is defined to incorporate these auxiliary tasks and the primary objective of language modeling. It serves as the optimization target during training and guides the model toward producing accurate and consistent responses.

$$\mathcal{L} = \mathcal{L}_{\text{LM}} + \alpha \mathcal{L}_{\text{aux}} \tag{1}$$

Here, $\alpha$ is a hyper-parameter that controls the trade-off between LM and the objectives of the auxiliary tasks.

## 3.2 Auxiliary Tasks

Recent research (Sankar et al., 2019) has shown that Transformer-based autoregressive models are robust to unrealistic perturbations at both the utterance and word levels. However, despite this robustness, the study suggests that these models have learned a bag-of-words-like representation rather than genuinely understanding language structure and meaning. On the other hand, understanding context is crucial to producing coherent and consistent responses in open-domain dialogue systems. While the connections between words within an individual utterance are essential for determining the meaning, it is also necessary to consider the relationships between utterances to fully understand the context of the conversation. To enhance the language model's comprehension and its ability to generate accurate and consistent outputs, it was deemed necessary to provide additional means to understand the relationships between the order of utterances and their meaning and to capture the sequential structure of language, as well as to comprehend the relationships between individual words in an utterance to grasp the semantic structure of language. We propose two auxiliary tasks for this purpose in this paper.

### 3.2.1 Utterance Permutation (UP)

In order to retain the sequential structure of language, re-ordering utterances is defined as an auxiliary generator in two ways: detection or recovering methods by rearranging 10% of utterances in a dialog chosen by 15% of all dialogues in the collection. Depending on the dataset, this task can be implemented based on a persona, context (history), or both.

In this work, we considered two approaches to implementing UP as auxiliary tasks:

- *detecting* (UPD), implemented as a binary token classification task.

- *recovering* (UPR), implemented as a non-binary token classification task.

In UPR, we attempt to predict the correct tokens regarding the re-ordered tokens; in UPD, we only determine whether or not the tokens are in the right place.

### 3.2.2 Utterance Masking (UM)

In our effort to comprehend the semantic structure underlying utterances, we devised the utterance masking task. This task is executed using two distinct approaches, analogously to the two methods described above:

- *detecting* the tokens in the masked utterances (UMD), implemented as a binary classification task.

- *recovering* the tokens in the masked utterance (UMR).

In both methods, 15% of the tokens within each dialogue are selected, with 80% of these tokens being replaced in the non-binary approach by the <mask> token and by synonyms in the binary approach. In the non-binary method, 10% of the tokens were randomly substituted from the dictionary, while in the binary approach, they are replaced with antonyms. The final 10% of tokens were preserved in their original form.

## 4 Dataset and Experiments

The following section provides detail on the dataset and experimental settings used in our experiments.

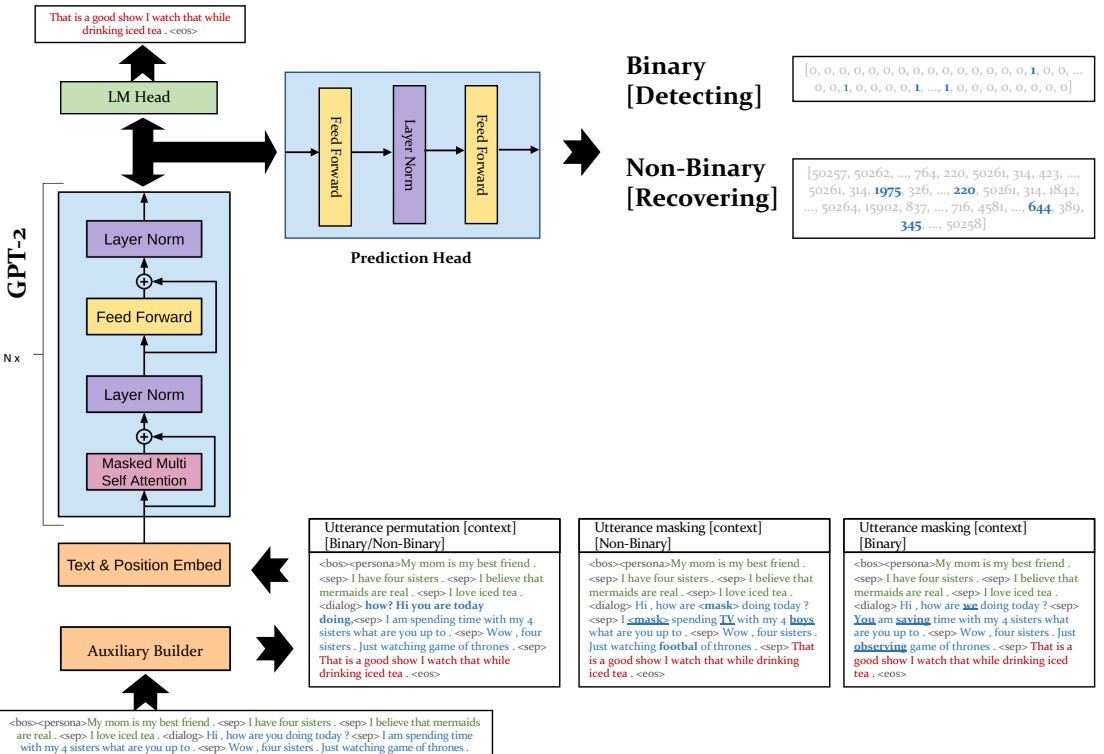

Figure 1: This figure illustrates the auxiliary tasks and the proposed model. The input to the model (our prompt) includes a combination of **persona**, **context**, and the **last conversation**. Each component is separated into special tokens and preceded by a unique token that signifies its component. The model's objective (LM objective) is to generate the final conversational component of the agent's response while disregarding any prior parts.

## 4.1 Dataset

The experiments in this paper are conducted using two benchmark datasets for open-domain dialogue generation, PersonaChat (Zhang et al., 2018) and DailyDialog (Li et al., 2017). PersonaChat is a large-scale dataset collected by encouraging two individuals to engage in open-domain conversations while exchanging personal information to create personas. The dataset contains over 163,064 utterances (11,907 dialogues) for training and 15,0264 utterances (968 dialogues) for testing. The conversations are naturally diverse, covering various topics and perspectives. In addition, the personal information provided allows the model to generate more informed and coherent responses due to predefined personalities. A true-cased version of PersonaChat[3] is used in the experiments to maintain consistency with the other datasets. On the other hand, the DailyDialog is a small dataset consisting of 13,118 multi-turn dialogues collected from various daily situations. Both datasets are pre-processed to ensure that all conversations are well-formed and coherent and that the data is in a

suitable format for training and the auxiliary generator. The datasets are split into training, validation, and test sets for experimentation.

## 4.2 Baselines

We compared our approach to DialoGPT (Zhang et al., 2020), a Transformer-based response generation model. We design a new prompt that is suitable for our auxiliary tasks. In order to ensure a fair comparison, we fine-tune the GPT-2 model on both of the two multi-turn datasets with the new prompt and using the same configurations introduced by DialoGPT (also known as VanillaGPT-2). This allows us to compare our approach to DialoGPT under the same conditions.

## 4.3 Implementation Details

Our implementation of both approaches is carried out using PyTorch Lightning[4] and Huggingface Transformers.[5] We train the baseline and our approach on two GPT-2 scales (small 124M and medium 354M parameters). Our approach depends

---

[3]bavard/personachat_truecased

[4]https://www.pytorchlightning.ai/
[5]https://huggingface.co/

on the dataset we implement the auxiliary tasks on different components persona, context, persona, and context, and by random. All the models are optimized with the AdamW optimizer (Loshchilov and Hutter, 2017) using an initial learning rate of 5e-5 and 3e-5, respectively, for the DailyDialog and PersonaChat datasets, and by using the adaptive learning rate scheduler with 5,000 warm-up steps and weight decay of 0.001. Experiments are performed on NVIDIA A100 for five epochs and a different range of hyperparameters regarding the auxiliary tasks, as seen in Table 1.

| Auxiliary Task | $\alpha$ | $P_{do}$ | $P_{reordered}$ | $P_{masked}$ | $P_{changed}$ |
|---|---|---|---|---|---|
| UPD | 3.0 | 0.15 | 0.1 | - | - |
| UPR | 1.0 | 0.15 | 0.1 | - | - |
| UMD | 3.0 | 0.15 | - | 0.8 | 0.5 |
| UMR | 1.0 | 0.15 | - | 0.8 | 0.5 |

Table 1: Hyperparameters used in the experiments.

### 4.4 Evaluation Metrics

The assessment of the models is performed in an automated manner utilizing well-established metrics such as perplexity (Vinyals and Le, 2015), BLEU (Papineni et al., 2002), and Rouge-L (Li et al., 2016a). In addition, we also incorporate two additional methods (similarity and correlation with human judgement) for automatic evaluation. These are the Embedding Average (Average), Embedding Extrema (Extrema), and Embedding Greedy (Greedy) metrics (Serban et al., 2017), which provide a deeper understanding of the correspondence between the model's responses and the reference responses. Furthermore, we compute the BertScore (Zhang et al., 2020) and MoverScore (Zhao et al., 2019) to assess the level of agreement between the generated text and human evaluations, and to determine the overall quality of the generated text.

### 5 Results

Table 2 presents the results of incorporating a combination of objectives and language modeling into various components of open-domain dialog systems. The evaluation was conducted on two benchmark datasets and two different scales of GPT-2. The results demonstrate that the improvement of the model depends on the type of auxiliary objective used in conjunction with language modeling. As demonstrated in the table, utilizing utterance permutation in binary form has a notable impact on reducing the perplexity of the model, with a reduction of 2% being observed.

Furthermore, compared to other auxiliary tasks, the use of utterance masking in the binary form leads to improvements in metrics such as BLEU, ROUGE-L, MoverScore, and Extrema. The results also suggest that using auxiliary tasks in larger models improves performance. The comparison between the Persona-Chat dataset highlights the significance of using auxiliary tasks simultaneously in both the Persona and Context components, which results in even better performance. Tables 3 and 4, located in Appendices A and B respectively, present sample generated responses for the two benchmarks, encompassing both the baseline and the optimal auxiliary model.

**What is the difference between binary and non-binary auxiliary tasks?** The results clearly demonstrate that the model only has access to the left context due to the specific type of attention mechanism employed in GPT (Masked Causal Attention). This limited exposure to context makes it challenging for the model to identify the distorted token correctly. Despite having access to the left context, the model's ability to recognize the scrambled token remains impaired.

**What is the impact of implementing these auxiliary tasks on different components of dialogue?** Determining the exact part of dialogue that will benefit the most from these tasks is challenging, but it can be agreed upon that combining both the Persona and Context components leads to improved outcomes.

**Why do the results vary across these two datasets?** The difference can be attributed to the distinct structures of the two benchmarks, as one provides access only to the context and the other to both persona and context.

**Does access to both persona and context result in higher quality answers?** This depends on the degree to which the persona aligns with the context.

### 6 Conclusion

In conclusion, our research has focused on improving the quality of generated responses using GPT-2 by proposing two auxiliary tasks. The first task, referred to as utterance permutation, aims to enhance the model's ability to comprehend the interconnections between words in a sentence and

DailyDialog

| Scale | Model | PPL | BLEU | ROUGE-L | BERTScore | MoverScore | Average | Greedy | Extrema |
|---|---|---|---|---|---|---|---|---|---|
| SMALL | VanillaGPT-2 | 11.463 | 1.188 | 0.187 | 0.885 | 0.045 | 0.875 | 0.737 | 0.881 |
| | UPD [context] | 11.445 | 1.108 | 0.186 | 0.884 | 0.042 | 0.873 | 0.735 | 0.877 |
| | UPR [context] | 11.607 | 0.984 | 0.179 | 0.883 | 0.038 | 0.870 | 0.734 | 0.880 |
| | UMD [context] | 11.484 | 1.365 | 0.188 | 0.885 | 0.047 | 0.870 | 0.736 | 0.881 |
| | UMR [context] | 11.859 | 0.999 | 0.184 | 0.884 | 0.040 | 0.871 | 0.735 | 0.879 |
| MEDIUM | VanillaGPT-2 | 10.344 | 2.603 | 0.208 | 0.889 | 0.072 | 0.880 | 0.743 | 0.881 |
| | UPD [context] | 9.958 | 2.393 | 0.203 | 0.888 | 0.064 | 0.881 | 0.745 | 0.883 |
| | UPR [context] | 10.192 | 2.068 | 0.199 | 0.887 | 0.060 | 0.877 | 0.739 | 0.882 |
| | UMD [context] | 10.659 | 2.458 | 0.208 | 0.889 | 0.075 | 0.879 | 0.745 | 0.882 |
| | UMR [context] | 10.554 | 1.886 | 0.195 | 0.886 | 0.055 | 0.874 | 0.740 | 0.880 |

PERSONA-CHAT

| Scale | Model | PPL | BLEU | ROUGE-L | BERTScore | MoverScore | Average | Greedy | Extrema |
|---|---|---|---|---|---|---|---|---|---|
| SMALL | VanillaGPT-2 | 13.149 | 1.489 | 0.099 | 0.879 | 0.056 | 0.878 | 0.694 | 0.872 |
| | UPD [persona] | 13.100 | 1.545 | 0.098 | 0.878 | 0.055 | 0.878 | 0.694 | 0.872 |
| | UPD [context] | 13.101 | 1.537 | 0.098 | 0.879 | 0.056 | 0.878 | 0.693 | 0.872 |
| | UPD [persona+context] | 13.089 | 1.426 | 0.097 | 0.878 | 0.054 | 0.877 | 0.693 | 0.872 |
| | UPD [random] | 13.108 | 1.552 | 0.097 | 0.878 | 0.055 | 0.877 | 0.693 | 0.872 |
| | UPR [persona] | 13.111 | 1.489 | 0.097 | 0.879 | 0.055 | 0.878 | 0.693 | 0.872 |
| | UPR [context] | 13.132 | 1.431 | 0.096 | 0.878 | 0.055 | 0.878 | 0.693 | 0.872 |
| | UPR [persona+context] | 13.125 | 1.586 | 0.097 | 0.879 | 0.055 | 0.878 | 0.694 | 0.872 |
| | UPR [random] | 13.128 | 1.427 | 0.097 | 0.878 | 0.054 | 0.877 | 0.694 | 0.872 |
| | UMD [persona] | 13.073 | 1.393 | 0.098 | 0.878 | 0.055 | 0.878 | 0.693 | 0.873 |
| | UMD [context] | 13.126 | 1.538 | 0.099 | 0.879 | 0.056 | 0.878 | 0.694 | 0.873 |
| | UMD [persona+context] | 13.079 | 1.504 | 0.097 | 0.878 | 0.054 | 0.878 | 0.692 | 0.872 |
| | UMD [random] | 13.055 | 1.423 | 0.096 | 0.878 | 0.055 | 0.877 | 0.693 | 0.872 |
| | UMR [persona] | 13.309 | 1.488 | 0.097 | 0.878 | 0.055 | 0.878 | 0.693 | 0.872 |
| | UMR [context] | 13.265 | 1.459 | 0.098 | 0.879 | 0.055 | 0.878 | 0.694 | 0.872 |
| | UMR [persona+context] | 13.362 | 1.371 | 0.096 | 0.878 | 0.053 | 0.878 | 0.693 | 0.872 |
| | UMR [random] | 13.263 | 1.454 | 0.098 | 0.878 | 0.055 | 0.878 | 0.694 | 0.872 |
| MEDIUM | VanillaGPT-2 | 10.975 | 1.657 | 0.100 | 0.879 | 0.060 | 0.878 | 0.695 | 0.873 |
| | UPD [persona] | 10.969 | 1.712 | 0.101 | 0.880 | 0.061 | 0.879 | 0.696 | 0.873 |
| | UPD [context] | 10.992 | 1.734 | 0.101 | 0.880 | 0.061 | 0.879 | 0.696 | 0.873 |
| | UPD [persona+context] | 10.960 | 1.693 | 0.101 | 0.879 | 0.060 | 0.879 | 0.695 | 0.873 |
| | UPD [random] | 10.978 | 1.690 | 0.101 | 0.879 | 0.060 | 0.878 | 0.694 | 0.872 |
| | UPR [persona] | 10.987 | 1.703 | 0.102 | 0.879 | 0.060 | 0.878 | 0.695 | 0.873 |
| | UPR [context] | 11.006 | 1.593 | 0.100 | 0.879 | 0.059 | 0.879 | 0.695 | 0.873 |
| | UPR [persona+context] | 11.000 | 1.660 | 0.100 | 0.879 | 0.060 | 0.879 | 0.695 | 0.873 |
| | UPR [random] | 11.004 | 1.575 | 0.099 | 0.879 | 0.059 | 0.879 | 0.695 | 0.873 |
| | UMD [persona] | 10.977 | 1.660 | 0.101 | 0.879 | 0.060 | 0.879 | 0.695 | 0.873 |
| | UMD [context] | 11.025 | 1.659 | 0.100 | 0.879 | 0.060 | 0.879 | 0.695 | 0.873 |
| | UMD [persona+context] | 11.004 | 1.714 | 0.101 | 0.879 | 0.060 | 0.879 | 0.696 | 0.873 |
| | UMD [random] | 10.957 | 1.593 | 0.099 | 0.879 | 0.059 | 0.879 | 0.694 | 0.872 |
| | UMR [persona] | 11.063 | 1.551 | 0.098 | 0.879 | 0.057 | 0.877 | 0.693 | 0.872 |
| | UMR [context] | 11.092 | 1.560 | 0.099 | 0.879 | 0.058 | 0.878 | 0.694 | 0.873 |
| | UMR [persona+context] | 11.102 | 1.468 | 0.099 | 0.879 | 0.057 | 0.878 | 0.694 | 0.873 |
| | UMR [random] | 11.044 | 1.540 | 0.099 | 0.879 | 0.058 | 0.879 | 0.695 | 0.873 |

Table 2: Results of the evaluation are based on automatic metrics. For each metric, colored numbers indicate the best-performing model.

produce grammatically accurate responses. The second task, utterance masking, is designed to improve the coherence and consistency of the generated responses by challenging the model to predict masked words based on the surrounding context.

Our experiments indicate that combining these two auxiliary tasks substantially improves the quality of generated responses. This includes improved grammar, coherence, and consistency, which are crucial aspects of generating high-quality NLP re-

sponses. Furthermore, these results demonstrate the potential of incorporating auxiliary tasks with language models to enhance their performance and improve the quality of generated responses in NLP applications.

## Acknowledgements

It is with deep gratitude that we thank the anonymous reviewers for their valuable feedback and guidance, which contributed significantly to the quality and clarity of our work. Finally, this research endeavor would not have been possible without the generous funding of the Wallenberg AI, Autonomous Systems and Software Program (WASP), funded by the Knut and Alice Wallenberg Foundation.

We would like to extend our appreciation to the Swedish National Infrastructure for Computing (SNIC), whose latest technologies enabled the computational aspects of our research. Furthermore, we acknowledge the Swedish Research Council's support for SNIC through grant agreement no. 2022/22-1003, which facilitated this research in part.

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

## A A comparison of Vanilla GPT-2 and GPT-2 with Auxiliary Tasks in Conversational Contexts for the DailyDialogue Dataset

| Model | Conversation | Ground response | Generated response |
|---|---|---|---|
| VanillaGPT-2 [sm] | - By the way miss, where is the toilet?
- Toilets are in the rear, I am afraid all the toilets are fully occupied at the moment.
- What? Oh, what we live! Thank you very much for your help, miss. | You are welcome. | - Thanks.
- I'll see you in Beijing in a few days.
- Goodbye, bye. |
| UPD [sm][context] | - By the way miss, where is the toilet?
- Toilets are in the rear, I am afraid all the toilets are fully occupied at the moment.
- What? Oh, what we live! Thank you very much for your help, miss. | You are welcome. | - Don't mention it.
- You are welcome.
- You are welcome. |
| VanillaGPT-2 [md] | - Yeah, look at all those new buildings going up! Mr. Zhang, the traffic is pretty smooth. But we were told the roads from the airport to downtown were quite crowded and traffic jams could be as long as half an hour.
- Yes, they were. But it has already past. The traffic from the airport to downtown has been relieved after the completion of Yan'an Aerial Road.
- We've never expected that Shanghai is such a big metropolis, same as New York city. David, would it be a good idea to rent a car here? I've got an international driver's license with a pretty good record in the States. | That's wonderful. OK. I'll remember that. But I have to warn you that you must drive very carefully in the downtown area, for there're no separate lanes for motor vehicles and bicycles in some places, and still quite a few people don't take road regulations seriously. | - Oh, yeah, you'd better have it.
- Oh, no, that's stupid. You'd better drive a taxi!
- That's a good idea. |
| UPD [md][context] | - Yeah, look at all those new buildings going up! Mr. Zhang, the traffic is pretty smooth. But we were told the roads from the airport to downtown were quite crowded and traffic jams could be as long as half an hour.
- Yes, they were. But it has already past. The traffic from the airport to downtown has been relieved after the completion of Yan'an Aerial Road.
- We've never expected that Shanghai is such a big metropolis, same as New York city. David, would it be a good idea to rent a car here? I've got an international driver's license with a pretty good record in the States. | That's wonderful. OK. I'll remember that. But I have to warn you that you must drive very carefully in the downtown area, for there're no separate lanes for motor vehicles and bicycles in some places, and still quite a few people don't take road regulations seriously. | - Mr. Zhang, I can assure you that the traffic isn't that bad here. There's a traffic island at the edge of the city to ease the traffic. But I still prefer to drive my own car.
- Mr. Zhang, I can assure you that the traffic isn't that bad here. There's a traffic island at the edge of the city to ease the traffic. But I still prefer to drive my own car.
- That would be great! |

Table 3: A comparison of Vanilla GPT-2 and Utterance Permutations (binary) models with small and medium scales, demonstrating ground truth responses and generated responses for various dialogues. Overall, the responses are more or less related to the context, but the GPT-2 with the auxiliary tasks produces responses that are more closely related to the context.

# B A comparison of Vanilla GPT-2 and GPT-2 with Auxiliary Tasks in Conversational Contexts for the PersonaChat Dataset

| Model | Persona | Conversation | Ground response | Generated responses |
|---|---|---|---|---|
| VanillaGPT-2 [sm] | - I love to drink wine and dance in the moonlight.
- I am very strong for my age.
- I'm 100 years old.
- I feel like I might live forever. | - Hi, how are you doing today?
- Good! you! Celebrating with fellow centenarians.
- Nice. I'm playing some card games with my family.
- That sounds like fun. I like wine and dancing too!
- Same. I try to get a small workout in a three mile walk for me is key.
- Awesome! Walking like that you might live forever like me!!
- That's true. My favorite way to unwind is reading a book. | You sound very like minded, just add some wine. | - I know! Where do you live? I'm still 100 years from retirement!
- What is your favorite book? Do you like to read?
- That's a big idea! I love books too. |
| VanillaGPT-2 [sm] | - I'm now looking for a new job.
- My favorite movie is Friday.
- I am a doctor.
- My daughter is a child prodigy. | - Hello, I'm sitting here with my dog. How are you? | I'm well friend. Looking for new employment at the moment. | - Hi my name is Dr. Robert.
- Hi how are you today.
- Hi. how are you doing? |
| UMB [sm][random] | - I love to drink wine and dance in the moonlight.
- I am very strong for my age.
- I'm 100 years old.
- I feel like I might live forever. | - Hi, how are you doing today?
- Good! you! Celebrating with fellow centenarians.
- Nice. I'm playing some card games with my family.
- That sounds like fun. I like wine and dancing too!
- Same. I try to get a small workout in a three mile walk for me is key.
- Awesome! Walking like that you might live forever like me!!
- That's true. My favorite way to unwind is reading a book. | You sound very like minded, just add some wine. | - That sounds like a lot of fun. Very healthy. I'm really strong. You should try to eat healthy! Books are fun. I have tons. |
| UMB [sm][random] | - I'm now looking for a new job.
- My favorite movie is Friday.
- I am a doctor.
- My daughter is a child prodigy. | - Hello, I'm sitting here with my dog. How are you? | I'm well friend. Looking for new employment at the moment. | - I'm doing good. Just finished my shift at the hospital. How are you?
- I'm doing well, and yourself?
- I am well, how about yourself. |
| VanillaGPT-2 [md] | - I prefer being inside.
- I am a recording engineer and own my own studio.
- Whats up I am a 46 year old single dad 46 a kids.
- I live in California but the recording artist market was dead last year. | - Hello, call of duty is my favorite game. Do you game? | Yes I do after all I am a engineer. | - Not as much as my daughter she's my baby.
- No I am just a big time recording engineer.
- I prefer the movie games and tv shows. I have a studio in California. |
| UMB [md][random] | - I prefer being inside.
- I am a recording engineer and own my own studio.
- Whats up I am a 46 year old single dad 46 a kids.
- I live in California but the recording artist market was dead last year. | - Hello, call of duty is my favorite game. Do you game? | Yes I do after all I am a engineer. | - I don't play game like that. I'm a live video coordinator.
- Not really. What do you do?
- I am not sure I've the computer but the tv show is my favorite. |

Table 4: An examination of Vanilla GPT-2 and Binary Utterance Masking (random in context and persona) models in small and medium sizes, showcasing authentic responses. The responses generally have a degree of contextual and persona relevance, but when the GPT-2 is integrated with auxiliary tasks, the responses demonstrate a stronger connection to the context and persona.