# OpenReview forum: "An Empirical Study of Multitask Learning to Improve Open Domain Dialogue Systems"
_NoDaLiDa/2023/Conference — NoDaLiDa 2023_

### Official Review · Reviewer_V4sH · 2023-03-08
**Good work**

**Rating:** 7
**Confidence:** 3

**Review:**

The paper considers the problem of generating responses in open-domain dialogue systems. Two auxiliary tasks are proposed to improve the quality of generated responses using GPT-2. Experimental results demonstrate the potential of incorporating auxiliary tasks with language models to improve the quality of generated responses.

Strengths
- Two auxiliary tasks are proposed (Utterance Permutation and Utterance Masking) in order to improve the quality of responses in dialogue systems.
- Experiments are carried out using two benchmark datasets (PersonaChat and DailyDialog).
- Two approaches  are used to implement auxiliary tasks (detecting and recovering).
- Evaluation metrics are described.

Weaknesses
- Further work is not designed.

**Paper Type:**

Long paper

---

### Official Review · Reviewer_k7V6 · 2023-03-13
**multitask leaning in decoder only models for open domain dialogue tasks**

**Rating:** 7
**Confidence:** 2

**Review:**

This paper presents a work on adding auxiliary tasks on decoder-only models for dialogue systems.
The authors proposed two auxiliary tasks, utterance permutation and utterance masking, implementing two methods each. Each method is added as another loss, training the decoder-only network in a multitask learning setup.
The authors evaluated the proposed model on two open-dialogue benchmarks, PersonaChat and DailyDialog, compared with a baseline model.

All in all the paper is focused on a clear objective. As an addition, it would also be nice to include the performance of a seq2seq model in the two benchmarks, even though it's out of the scope but to give a full picture of the performance of another model type for the same tasks. Finally, some output of the different models could be shown, as qualitative analysis, to see the differences in the output of the models.

In the abstract there is an error, line 11, it should be "decoder-only".



**Paper Type:**

Short paper

---

### Decision · Program_Chairs · 2023-03-17

Accept